# Novel COVID-19 Diagnosis Delivery App Using Computed Tomography Images Analyzed with Saliency-Preprocessing and Deep Learning

**Santiago Tello-Mijares** [1,*] and **Fomuy Woo** [2]

1 Postgraduate Department, Higher Technological Institute of Lerdo,
National Technological Institute of Mexico Campus Lerdo, Lerdo 35150, Mexico

2 Medical Family Unit, Institute of Security and Social Services for State Workers, Torreon 27268, Mexico;
luisa.fomuy@issste.gob.mx

* Correspondence: jtello@itslerdo.edu.mx; Tel.: +52-8712210810

**Abstract:** This app project was aimed to remotely deliver diagnoses and disease-progression information to COVID-19 patients to help minimize risk during this and future pandemics. Data collected from chest computed tomography (CT) scans of COVID-19-infected patients were shared through the app. In this article, we focused on image preprocessing techniques to identify and highlight areas with ground glass opacity (GGO) and pulmonary infiltrates (PIs) in CT image sequences of COVID-19 cases. Convolutional neural networks (CNNs) were used to classify the disease progression of pneumonia. Each GGO and PI pattern was highlighted with saliency map fusion, and the resulting map was used to train and test a CNN classification scheme with three classes. In addition to patients, this information was shared between the respiratory triage/radiologist and the COVID-19 multidisciplinary teams with the application so that the severity of the disease could be understood through CT and medical diagnosis. The three-class, disease-level COVID-19 classification results exhibited a macro-precision of more than 94.89% in a two-fold cross-validation. Both the segmentation and classification results were comparable to those made by a medical specialist.

**Keywords:** artificial intelligence; computed tomography; COVID-19; SARS-CoV-2; medical diagnostic imaging

## 1. Introduction

COVID-19 is an infectious disease with continuously emerging variants [1,2]. Real-time polymerase chain reaction (RT-PCR) is the principal technique used for the early diagnosis of COVID-19 infection [3,4]. Thereafter, chest computed tomography (CT) assumes an important role in assessing the extent of the lung damage in patients with moderate-to-severe COVID-19 pneumonic disease [4].

CT is one of the main techniques used to assess the severity of a pneumonic infection [5–10]. This approach allows patients to be stratified into risk categories, provides prognosis estimates, and facilitates informed medical decision making [10]. The most common CT finding associated with clinical severity is the extent of pulmonary involvement and pneumonia. [5,6,8,9,11,12]. The symptoms of severe COVID-19 are manifestations of ground glass opacity (GGO) and pulmonary infiltrates (PIs) in the peripheral subpleural region visualized on CT images; these symptoms usually occur within 14 days of exposure to the virus [13].

During the pandemic, CT-based evaluation became increasingly relevant for treating critically ill patients with COVID-19 who were seriously ill. CT demonstrated findings classified as common in COVID-19 pneumonia, according to the recommendations of the Radiologic Society of North America [14]. CT also plays a fundamental role in intensive

care units for detecting and monitoring life-threatening complications related to COVID-19. Using CT improves prognosis and survival predictions in patients with COVID-19 [15].

Deep learning can be used to perform a quantitative and rigorous analysis of the infected volume on CT images [16,17]. Deep learning enables computers to perform human tasks [18,19]. Information is available on representative deep learning applications against the COVID-19 pandemic [20]. The authors of [21–26] suggested using different artificial intelligence techniques to detect and diagnose COVID-19 using CT images, which was the aim of the present study.

Mobiny et al. [21] presented a deep learning technique called DECAPS (detail-oriented capsule networks) to classify COVID-19 from CT images. Amyar et al. [22] proposed a deep learning model to segment and identify a COVID-19-infected region from CT images. Polsinelli et al. [23] used a SqueezeNet based-model to classify COVID-19 CT images. Li et al. [24] extracted features from 50 layers of a neural network (COVNet) to classify COVID-19 on CT images. Debanjan Konar et al. [25] used a parallel quantum-inspired self-supervised network (PQIS-Net)-assisted semi-supervised shallow learning method for COVID-19 classification from CT images. Yu-Huan Wu et al. [26] proposed a JCS (joint classification and segmentation) system to classify and diagnose COVID-19 using CT images.

This research builds on those works as the next step by using image indicators and classification to categorize the damage and classify the progression of pneumonia over the pulmonary parenchyma (PP) in patients with COVID-19 (Figure 1a–c). In this paper, we present a competitive method for identifying the progression of pneumonia with the saliency of GGO and PI pattern areas and a deep learning technique for three-class classifications (Figure 1d–f).

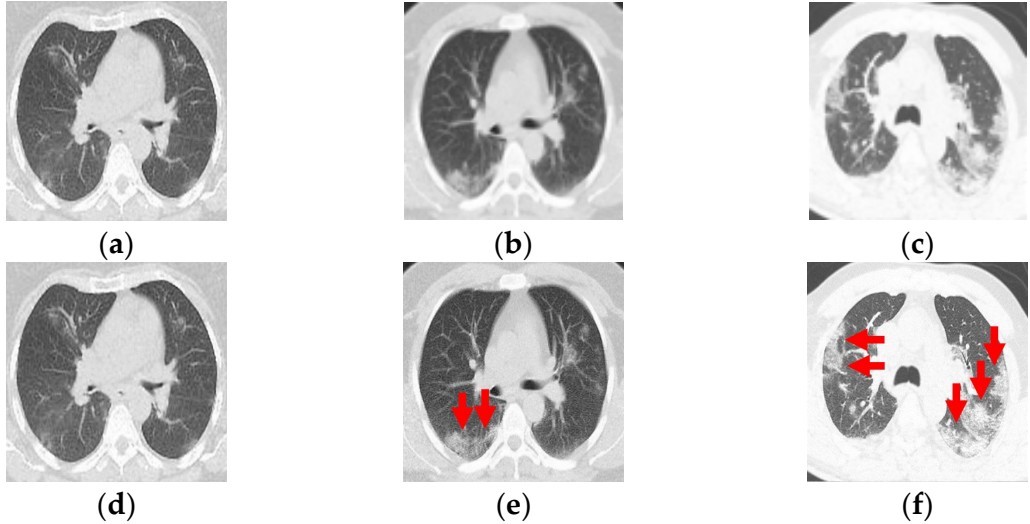

**Figure 1.** (**a**–**c**) Computed tomography from a patient with COVID-19. (**d**–**f**) Ground-glass opacity identification (red arrow marking) for three-class damage progression classification: (**d**) Class 1, (**e**) Class 2, and (**f**) Class 3.

The paper is organized as follows. Section 2 describes the proposed GGO and PI preprocessing saliency and the three-class classification deep learning model for CT images (Figure 2). Section 3 presents the classification results obtained in two-fold cross-validation. Section 4 describes and discusses the results before finally concluding the paper.

In this work, we devised a web application in MATLAB (MathWorks, Natick, MA, USA) designed to optimize shared workflows between doctors. Image processing and artificial intelligence were used to classify and diagnose the severity level of COVID-19. This system will help with the rapid hospitalization of high-risk patients while also preventing the passing of documents from hand to hand.

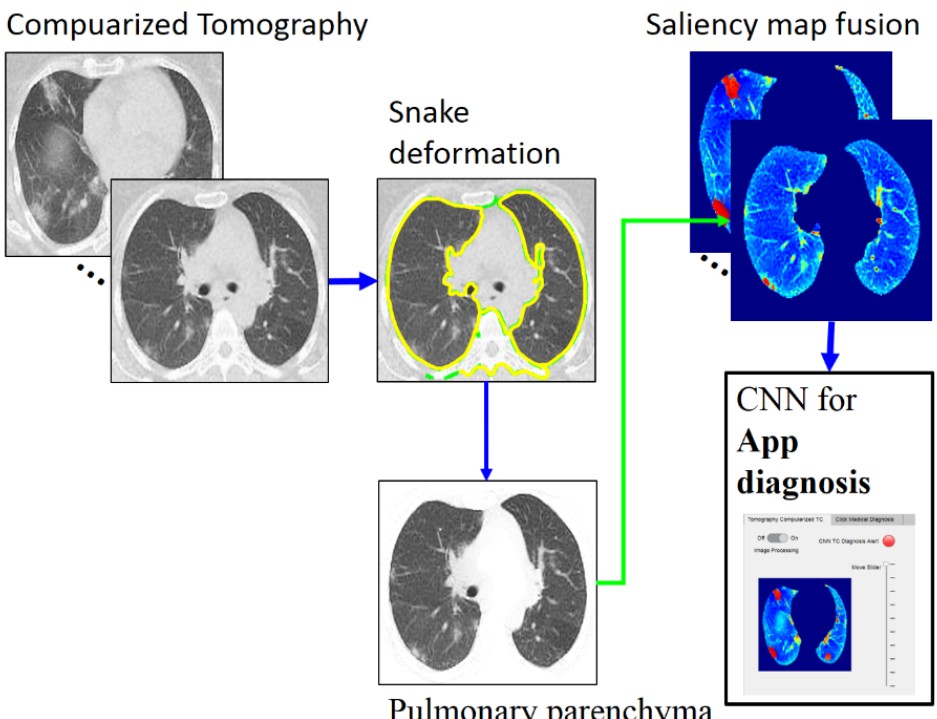

**Figure 2.** Overall method description, https://drive.google.com/drive/folders/1Q41lGFpawPvYmPZ8 SNK3sgsuvjs4kB45?usp=sharing (accessed on 3 March 2022).

## 2. Materials and Methods

Figure 2 depicts the phases of the proposed classification method, which are detailed in the following subsections. The proposed method is accomplished in three stages. The first is PP image segmentation using an automatic active model (ACM) [27] via a Poisson inverse gradient (PIG) [28] based on our previous work [29]. Next is applying saliency maps to highlight the GGOs and PIs. Finally, CNN-based classification (three-class disease classification) is performed using the web app in MATLAB. The CT images used for the present analysis were from 54 RT-PCR-confirmed COVID-19 patients with "typical" disease presentation (GGO, PI, interlobular thickening, crazy-paving pattern, consolidation, reverse halo sign, and double reverse halo sign) [30]; all included 54 patients underwent RT-PCR confirmation for COVID-19.

### 2.1. Pulmonary Parenchymal Identification by Poisson Inverse Gradient

The CT images were converted to BMP files for image processing and data analysis using MATLAB. Based on our previous work [29], we used an ACM (active contour model) [27] in which the energy function (*E*) is minimized to identify the contours of the PP (Figure 3a). We used the ACM vector field by applying an active model via PIG (Poisson inverse gradient) [28]. The PIG method estimates the energy field (*E*) such that the negative gradient of *E* is the vector field closest to the force field *F* in the *L2-norm* sense:

$$E(v) = \int_0^1 \int_0^1 [E_{\text{int}}(v) + E_{ext}(v)]ds \tag{1}$$

where *v(s)* is a surface; *v(s) = (x(s), y(s))* and $E_{int}$ and $E_{ext}$ are the internal and external convolution (VFC) method energies of the image, respectively. The internal energy is defined as:

$$E_{\text{int}} = \frac{1}{2}\left(\alpha \sum_s |v_s|^2 + \beta \sum_s |v_{ss}|^2\right) \tag{2}$$

where $v_s$ and $v_{ss}$ are the first and second derivatives of the surface v(s), respectively, and $\alpha$ and $\beta$ are weighting parameters to constrain the degree of elasticity and rigidity of the surface, respectively. For our experiments, we used an implementation [28] where $\alpha = 0.5$ and $\beta = 0.1$.

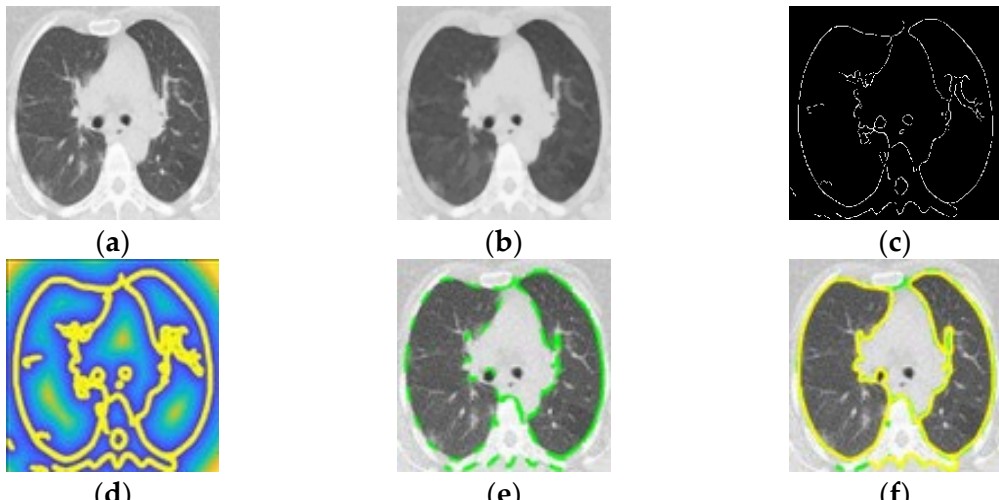

**Figure 3.** (**a**) Computed tomography in patient with COVID-19. (**b**) Smoothed image. (**c**) Optimized edge map B. (**d**) External energy of the VFC field superposed with isolines. (**e**) PIG initialization. (**f**) Final snake deformation.

The edge force is the gradient of the image's edge map B. $E_{ext}$ was calculated using the optimized edge map *B* using a Canny edge detector (Figure 3c) [31] from the smoothed image (Figure 3b). The external energy represents the image's edge information, which is defined as:

$$E_{ext}(v) = -|\nabla(G_\sigma(x,y,z) * I(x,y,z))|^2 \qquad (3)$$

where $G_\sigma$ and *I* are the Gaussian function and the image, respectively, and $\nabla$ is the gradient operator. Applying the VFC algorithm provided a smooth contour (Figure 3d) for the inner cross-section within 40 iterations (initial iteration in Figure 3e and final iteration in Figure 3f). Finally, the inner cross-sectional area of the PP was calculated by integrating the detected contour in a segmented area of the PP (Figure 4a).

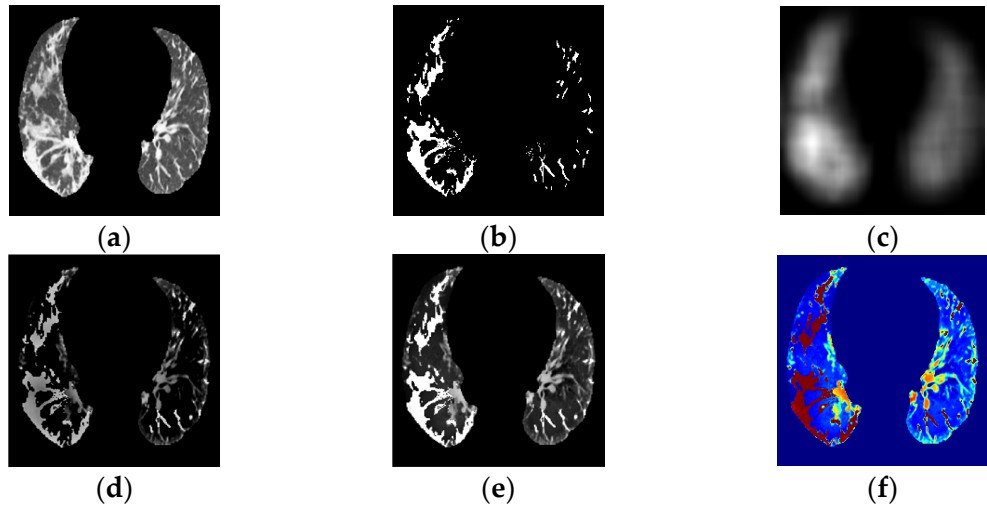

**Figure 4.** (**a**) PP area image. (**b**) GGO–PI image. (**c**) Base fused image. (**d**) Detail fused image. (**e**) Fused image. (**f**) Transformed RGBjet Fused image.

## 2.2. Ground-Glass Opacity and Pulmonary Infiltrates Highlighted by Saliency Fusion

This subsection describes a segmentation and fusion process. An initial PP area identification (Figure 4a) and GGO–PI candidate-region selection (Figure 4b), based on the application described in our previous work [29], were followed by saliency fusion.

To identify ground-glass opacity (GGO) and pulmonary infiltrates (PIs) on segmented areas of PP, we used an efficient technique that used over-segmentation by mean shift and superpixel-SLIC (simple linear iterative clustering) on a CT image. First, by applying watershed segmentation to the mean-shift clusters, only PP segmentation-identified zones showed GGO and PIs based on the description of each watershed cluster regarding its position, grey intensity, gradient entropy, second-order texture, Euclidean position to the border region of the PI zone, and global saliency features after using TRF (tree random forest).

The objective was to isolate or highlight all regions that might be GGOs and PIs and to fuse these regions and the PP area for highlighted saliency identification.

The next step was PP and GGO–PI image fusion. Image fusion involves integrating relevant, complementary, and redundant visual information from multiple sources into a single composite image [32]. From the PP (Figure 4a) and GGO–PI (Figure 4b) images, the base fused image (Figure 4c) was extracted by combining the mean filter from both images; second, the detailed fused image (Figure 4d) was obtained by combining the median filter and the weight maps of visual saliency from both images. Last, the saliency fused image was obtained by calculating these two images (Figure 4e).

The principal contribution in this section is the saliency fusion approach for the GGOs and PIs highlighted in the PP on CT images.

The visual saliency process was used to obtain the most relevant information within the PP area to highlight identified GGO-PI, and obtain the final fusion image results using a weight-map approach (Figure 4e). This method highlighted the pertinent information in both images and combined these into a final transformed RGBjet fused image (Figure 4f) to feed to the CNN.

## 2.3. Convolutional Neural Network Classification for Telemedicine App

The resulting fused images (Figure 5d–f) allowed us to diagnose the evolution of COVID-19 and grade the damage caused by the infection using a CNN. The literature contains up to four categories for the standardized reporting of COVID-19-related chest CT findings proposed by the RSNA Expert Consensus Statement [33]. In addition, the prevalence of this COVID-19 CT pattern did not statistically differ over the course of the two-year pandemic, and this new type of pneumonia tends to present with "typical" radiological features in most patients [30].

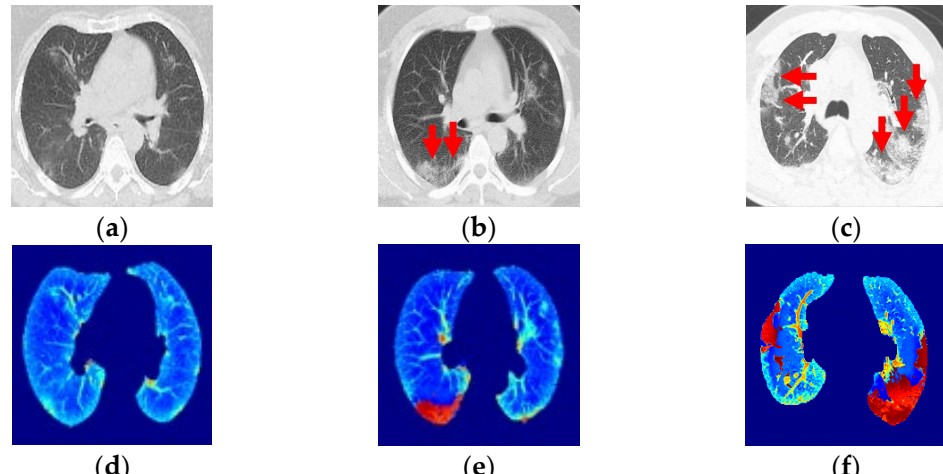

**Figure 5.** (**a**,**b**) Ground-glass opacity identification (red arrow marking) for three-class damage progression classification: (**a**) Early phase. (**b**) Progression phase. (**c**) Peak phase. (**d**) Class 1, early phase. (**e**) Class 2, progression phase. (**f**) Class 3, peak phase.

This section presents the application of a CNN to categorize clinical CT images into three categories based on the relationship between CT infection findings, symptoms, and symptom evolution [34–36]. The three evolutionary phases starting from the onset of symptoms are early, progression, and peak.

Class 1, the early phase (Figure 5a, 0 to 4 days): Slightly predominant GGO pattern and CT in 50% of the first two days may appear normal.

Class 2, the progression phase (Figure 5b, 5 to 8 days): Infection with GGO and PIs rapidly spreads and becomes bilateral. Patterns in the infected regions may appear as cobblestones and consolidations.

Class 3, the peak phase (Figure 5c, 9 to 14 days): GGO regions change to predominantly consolidation and some cobblestone regions.

The images were categorized into these three classes, with Class 3 being the final, most critical phase with the maximum peak.

A CNN, as originally proposed by Yan LeCun et al. [37], is a deep learning neural network model with three key architectural ideas (local receptive fields, weight sharing, and sub-sampling in the spatial domain), and it consists of three main types of layers: spatial convolution layers (CONV l), subsampling pooling layers (MaxPool l), and fully connected layers (FC l).

We used a CNN based on previous studies designed to process three-dimensional (3D) images: AlexNet [38]. A CNN is optimal for feature extraction because it is hierarchical (with multiple layers for greater compactness and efficiency) and not variance-redundant (regarding position, size, luminance, rotation, pose-angle, noise, and distortion). Many comprehensive reviews of CNNs are available in the literature [39,40].

The proposed method extracts GGO–PI characteristics for classification features from the saliency image using the CNN. Our main contributions are generating input images and capturing relevant GGO–PI information to train and feed the CNN.

CNNs generally operate on thousands of standard-size input images that require prior adaptation. The advancements explored in this phase include replacing this adaptation process with a transformed RGBjet fused image. For each input CT image, transformed RGBjet fused images were generated. The images were first resized to 227 × 227 × 3 according to the AlexNet training set [41]. These RGB images (227 × 227 pixels × three color levels) were fed to the CNN (Figure 6), which was designed to extract 1043 high-level feature vectors, obtained by passing through the layers of the AlexNet architecture [41], using $O = ((I - K + 2P)/S) + 1$ for CONV layers and $O = ((I - Ps)/S) + 1$ for MaxPool layers, where O is output image size in both cases and depending on the following variables, I is the size of the input image, K is the width of the kernels (filters) used in the convolution layer, N is the number of kernels, S is the stride (pixel jumps) of the convolution operation, P is padding (fill-no), and Ps is the MaxPool filter size.

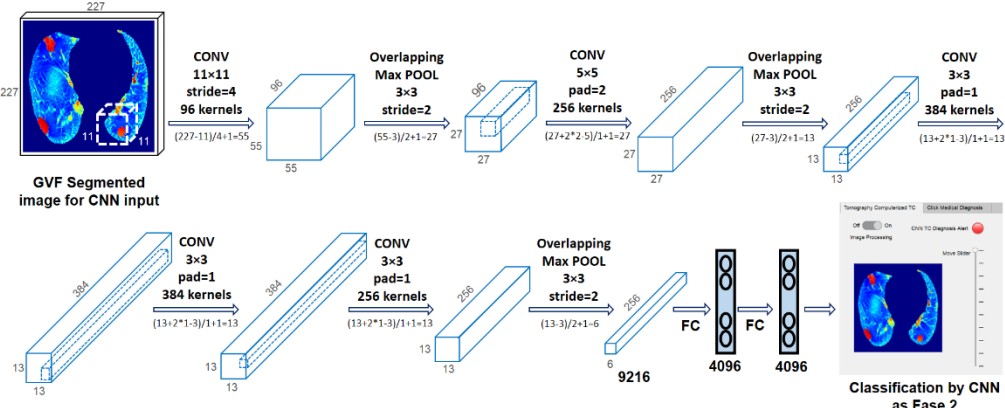

**Figure 6.** Convolutional neural network [41], https://drive.google.com/drive/folders/1Q4 1lGFpawPvYmPZ8SNK3sgsuvjs4kB45?usp=sharing (accessed on 3 March 2022).

Figure 6 shows the evolution of the $227 \times 227 \times 3$ input image through the different layers of the CNN. After CONV 1 (96 $11 \times 11$ filters), the size changed to $55 \times 55 \times 96$, which became $27 \times 27 \times 96$ after MaxPool 1 ($3 \times 3$ pooling). After CONV 2 (256 $5 \times 5$ filters), the size changed to $27 \times 27 \times 256$, and then MaxPool 2 ($3 \times 3$ pooling) changed it to $13 \times 13 \times 256$. CONV 3 (384 $5 \times 5$ filters) transformed it to $13 \times 13 \times 384$, CONV 4 (384 $5 \times 5$ filters) preserved the size, and CONV 5 (256 $5 \times 5$ filters) changed it back to $27 \times 27 \times 256$. Finally, MaxPool 3 ($5 \times 5$ pool) reduced the size to $6 \times 6 \times 256$. This image was fed to FC 1, transforming it into a $4096 \times 1$ vector. These vectors contained the features of the CNN extraction for classification.

We used two-fold cross-validation, in which we randomly shuffled the dataset into two sets, $d_0$ and $d_1$, so that both sets were the same size. We first trained on $d_0$ and validated on $d_1$, followed by training on $d_1$ and validating on $d_0$.

The image processing results, CNN classification, and medical diagnosis on the hospital admission sheet were included in the MATLAB web app, facilitating streamlined workflow sharing. In this work, telemedicine was used as a virtual care platform that allowed for communication between respiratory triage/radiology and the COVID-19 multidisciplinary teams (Figure 7). It could be an essential medical instrument for containing the spread of COVID-19 and managing the care for COVID-19 patients in need of beds and further hospitalization.

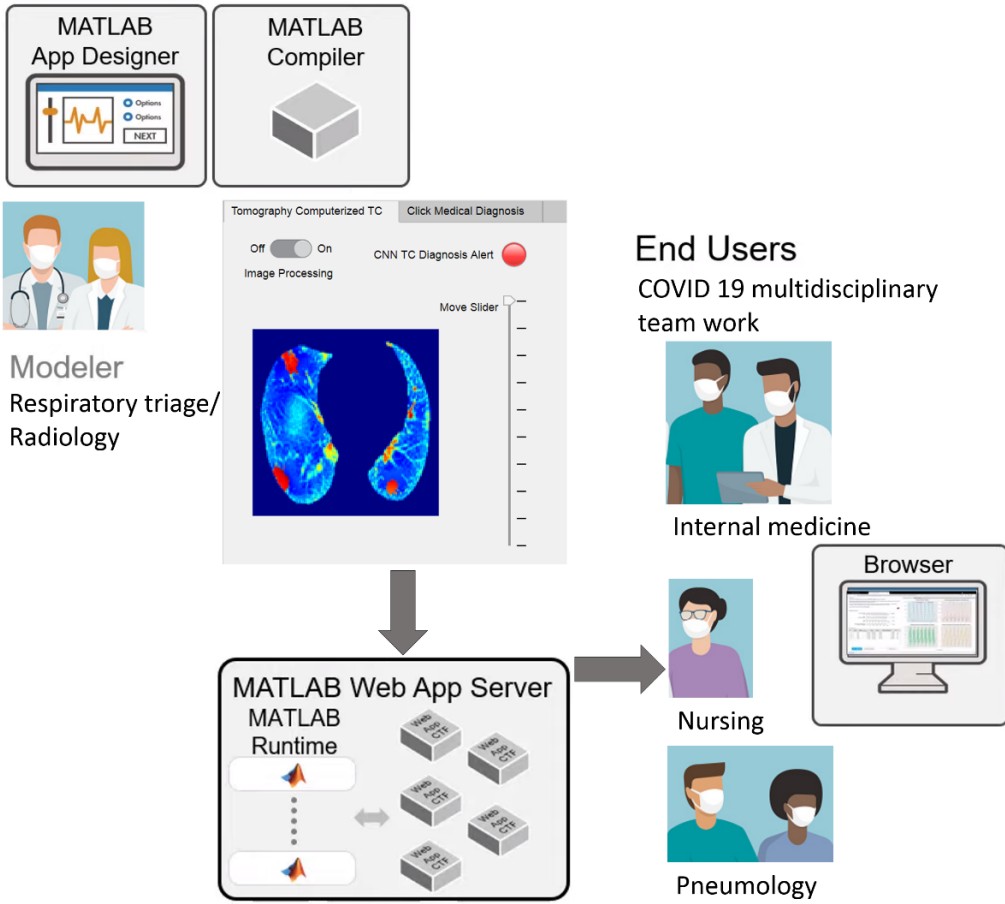

**Figure 7.** Schematic of streamlined telemedicine workflows, https://drive.google.com/drive/folders/1Q41lGFpawPvYmPZ8SNK3sgsuvjs4kB45?usp=sharing (accessed on 3 March 2022).

Hospital admissions are reserved for patients with urgent needs based on their symptoms. A patient arrives at the first contact physician (emergency physician or general physician) in respiratory triage, who, upon finding evidence of pulmonary compromise or respiratory distress, requests a CT from the radiology service. The radiologist assesses

the CT image and, for cases with lung or lower-respiratory-tract involvement, refers the patient through the "referral sheet" to the second level of care. At this hospital, the CT is taken with a Philips Brilliance 16-slice helical CT scanner (Koninklijke Philips, Eindhoven, NV, Nederland).

Through the web app, the CT is sent with the CNN diagnosis and the reference sheet instead of being transferred hand-to-hand. The dated reference sheet includes the patient's identifying information, personal history, current condition, established treatments, laboratory and imaging reports, and the diagnosis or reason for referral. Telemedicine describes a virtual care platform that enables remote diagnosis using telecommunication technology and, as such, represents an indispensable medical instrument in reducing COVID-19 spread [42,43].

## 3. Experimental Results

### 3.1. Dataset

The used dataset included 540 images (420 × 280 pixels) in the BMP format from 54 patients (10 CT images each). The ground-truth data (i.e., the identification of PP and their degree of alteration) were generated by a specialized doctor based on the GGO–PI localization.

### 3.2. Quantitative and Qualitative Evaluation of PP and GGO–PI Identification

To assess the segmentation quality, we compared the PP region with a correctly identified corresponding region in the ground truth. This comparison was quantified using the Zijdenbos similarity index [44], $ZSI = 2*A1 \cap A1 / |A1| + |A2|$, where A1 and A2 refer to the compared regions (both binary masks). A ZSI value greater than 0.75% was considered to represent excellent agreement [44]. The PP identification results had a ZSI of $0.9484 \pm 0.0030\%$, which indicates that the obtained segmentation agreed with the ground truth. This proves that the proposed segmentation approach can obtain results similar to those manually obtained by a medical expert. For GGO, the PI identification showed 96% precision and 96% recall in the two-fold cross-validation.

### 3.3. Quantitative Indicators for CNN Diagnosis

To quantitatively assess the three-class classification results and the performance of the CNN technique, several quality indicators were obtained. We used final or external quality indicators to evaluate the final classification results and to enable external comparison with other works. For multiclass problems, we used precision, recall, and F1 score metrics in a specific way. Let TP, FP, and FN be the number of true positives, false positives, and false negatives, respectively. For these metrics to be calculated in a multi-class problem, the problem needed to be treated as a set of binary problems ("one vs. all"). We then defined the precision or positive predictive value (PPV = TP/TP + FP); sensitivity, recall, or true positive rate (TPR = TP/P); and F1 score (F1 = 2*PPV*TPR/ PPV+TPR).

In this case, a metric could be calculated per class, and then the final metric was the average of the per-class metrics. These three metrics were calculated three times (Table 1). From the confusion matrix results (Figure 8), for each class, we determined the TP, FP, and FN by treating it as a "one vs. all" problem. Let P1 = 252 be the number of Class 1 cases in the dataset, P2 = 109 for Class 2, and P3 = 179 for Class 3. After conducting the same process for all of the classes, we collated and summed these values in Table 1. Next, we calculated the precision, recall, and F1 scores for each class.

Based on Table 1, the macro-average precision was calculated as the average precision of all classes ($PPV_{macro} = (PPV_1 + PPV_2 + PPV_3)/3$); the macro-average recall was calculated as $TPR_{macro} = (TPR_1 + TPR_2 + TPR_3)/3$; and the macro-average F1 score was calculated as $F1_{macro} = (F1_1 + F1_2 + F1_3)/3$. These results are presented in Table 2.

**Table 1.** Metrics for quantitative indicators from the two-fold cross-validation.

|  | TP | FP | FN | P | $n_s$ | PPV | TPR | F1 |
|---|---|---|---|---|---|---|---|---|
| Class 1 | 245 | 7 | 7 | 252 | 259 | 0.9722 | 0.9459 | 0.9589 |
| Class 2 | 99 | 10 | 10 | 109 | 119 | 0.9082 | 0.8319 | 0.8684 |
| Class 3 | 173 | 6 | 6 | 179 | 185 | 0.9664 | 0.9675 | 0.9505 |
| TOTAL | 517 | 23 | 23 |  |  |  |  |  |

F1, F1 score; FN, false negatives; FP, false positives; ns, number of simple; P, number of class 1, 2, or 3 cases in the dataset; PPV, precision or positive predictive value; TP, true positives; and TPR, sensitivity, recall, or true positive rate.

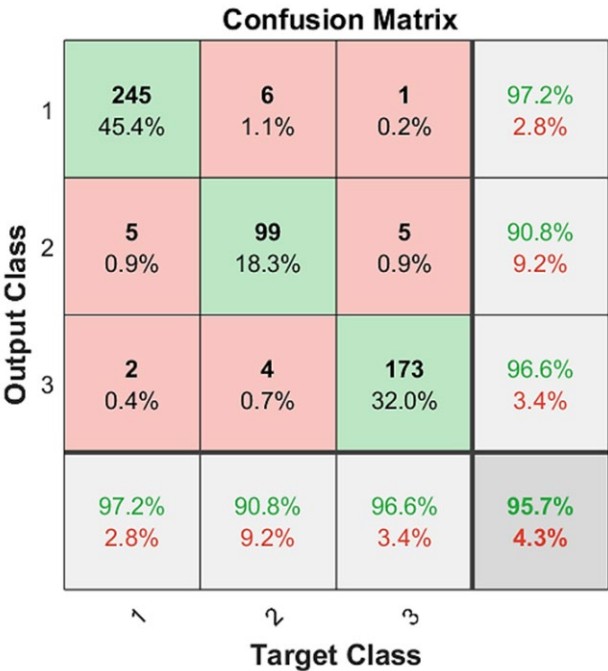

**Figure 8.** CNN classification results on two-fold cross validation: green as correctly classified instances and red as incorrectly classified instances, https://drive.google.com/drive/folders/1Q4 1lGFpawPvYmPZ8SNK3sgsuvjs4kB45?usp=sharing (accessed on 3 March 2022).

**Table 2.** Quantitative results on two-fold cross-validation.

|  | PPV | TPR | F1 |
|---|---|---|---|
| Macro | 0.9489 | 0.9043 | 0.9259 |

F1, F1 score; PPV, precision or positive predictive value; and TPR, sensitivity, recall, or true positive rate.

Macro-averages are preferable for three-class CT diagnosis problems because these problems present a class-imbalanced dataset where one class is more important than the two others. Class 3 is the most important because it contains the patients in the most critical condition. Additionally, macro-averages are preferable because they highlight a model's performance across all classes equally.

## 4. Discussion

Table 3 presents the best quality indicators for the state-of-the-art COVID-19 diagnosis classification methods.

Most of the deep learning publications on COVID-19 concentrate on the classification of COVID-19 or non-COVID-19. Alternatively, RT-PCR results can be used to perform this classification.

**Table 3.** Comparison of COVID-19 diagnosis classification methods (%).

| Reference | Classes | Set Images | Training/Test | PPV | TPR | F1 |
|---|---|---|---|---|---|---|
| Proposed Vgg19 | Early, Progression, Peak | 540 (252, 09, 179) | 2-fold class validation | 0.948 | 0.904 | 0.925 |
| [21] | COVID-19, non-COVID-19 | 746 (333, 397) | 0.85/0.25 | 0.843 | 0.915 | |
| [22] | COVID-19, non-COVID-19 | 1044 (449, 595) | 944/100 | | 0.81 | 0.83 |
| [26] | COVID-19, non-COVID-19 | 757 (360, 397) | 0.6 train/0.2 validation/0.2 test | 0.8173 | 0.85 | 0. 83 |
| [24] | COVID-19, community-acquired pneumonia, non-pneumonia | 4352 (1292, 1735, 1325) | 0.9/0.1 | | 0.9 | |
| [25] | COVID-19, non-COVID-19 | 2482 (1252, 1230) | 1764/718 | 0.965 | 0.935 | 0.948 |
| [26] | COVID-19, non-COVID-19 | 144,167 (64,711, 68,041) | 2794–7500/64, 711–68,041 | | 0.95 | |

F1—F1 score; PPV—precision or positive predictive value; TPR—sensitivity, recall or true positive rate.

In this work, CT was used for patient hospitalization based on the progress of the disease (early, progression, or peak). However, our study was limited by the small size of the true training and test sets that therefore may not represent the entire patient population.

We compared the results of our work with published methods based on deep learning developments [21–26]. Although the methods and datasets (training and testing) used here and elsewhere differed, we compare respective results, as summarized in Table 3. The methods derived in [25,26] achieved better results than ours but only [26] classified patients into three classes, and none employed two-fold cross-validation.

Despite our dataset being relatively sparse, we outperformed radiologists on a high-quality subset of test data. More importantly, a respiratory triage/radiologist and a COVID-19 multidisciplinary team used our model to review and revise their clinical decisions, which suggests the importance of this promising clinical visualization instrument.

In addition, the proposed deep learning classifier obtained three-class classification results unlike other studies regarding the diagnosis of COVID-19 abnormalities. To some extent, these comparisons and our classification results confirm that the studied CNN approach is valid; for a dataset containing 540 COVID-19 images (252 early phase, 109 progression phase, and 179 peak phase), we obtained values of PPV = 0.9489, TPR = 0.9043, and F1 = 0.9259 using the most stringent two-fold cross-validation compared to other similar "state-of-the-art" studies.

## 5. Conclusions

The identification of the PP in CT images is generally accepted as a prerequisite to aid in the segmentation of lung regions and extract features for candidate infection-region detection or COVID-19 classification (healthy or infected). In this study, we proposed a method based on PIG to identify and segment the PP in CT images in practical situations using a MATLAB app. We obtained identification results with a ZSI of $0.9483 \pm 0.0031\%$ and segmentation results comparable to those obtained by a medical specialist in a representative dataset with a precision of 96% and a recall of 96% in two-fold cross-validation for GGO and PI identification.

To classify the CT images for disease diagnosis in COVID-19 patients, our main contributions were the novel use of over-segmentation and a GGO-highlighted fusion saliency technique for the precise CNN classification of segmented regions into three classes (early, progression, and peak). We obtained classification results with a macro-precision of

more than 94.89% in a two-fold cross-validation. We thoroughly analyzed reported works on infection-region detection and COVID-19 classification. We demonstrated why the model presented here represents the next logical step in progressing the field of automated CT image analysis (application for telemedicine). We discussed our results while assessing the novelty and quality of our achievements in the absence of a systematic framework to objectively compare our results.

Furthermore, we presented an app based on the MATLAB platform. This interface includes CT and image processing analysis, and it presents GGO–PI with the highlighted identification of the PP by saliency map fusion and CNN classification results (early, progression, or peak phase). In addition, the reference sheet was digitized and no longer requires being passed along hand-to-hand.

**Author Contributions:** Conceptualization, methodology, formal analysis, investigation, writing—original draft preparation, writing—review and editing, visualization, supervision, and project administration, S.T.-M. and F.W.; software and resources, S.T.-M.; validation and data curation, F.W.; all authors have read and agreed to the published version of the manuscript. All authors have read and agreed to the published version of the manuscript.

**Funding:** This research received no external funding.

**Institutional Review Board Statement:** Not applicable.

**Informed Consent Statement:** Not applicable.

**Data Availability Statement:** The data and materials are available for research purposes.

**Acknowledgments:** This work was partially supported by Internal Grant "ITSLerdo #001-POSG-2020". This study was partially supported by Internal Grant no. 312304 from the F0005-2020-1 CONACYT Support for Scientific Research Projects, Technological Development, and Innovation in Health in the Face of Contingency by COVID-19.312304 from the F0005-2020-1.

**Conflicts of Interest:** The authors declare no conflict of interest.

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
