# Peer review of "Novel COVID-19 Diagnosis Delivery App Using Computed Tomography Images Analyzed with Saliency-Preprocessing and Deep Learning"

_tomography, doi:10.3390/tomography8030134_

Round 1

Reviewer 1 Report

SPECIFIC COMMENTS

This manuscript is pleasurable and flowing; no major issues are appreciable.

Although language used is appropriate, I (I am not a native English speaker) recommend to Authors to obtain a certified native speaker with proficiencies in the scientific-medical field to complete properly this paper (if not jet done). Moreover, I recommend making a further revision of the manuscript to fix some small typing/language errors.

TITLE

The title is clear and direct. Personally, I believe it could be improved and be more focused on results. “Saliency-preprocessing for covid-19 diagnosis on computed tomography by Deep Learning”. Moreover, I believe that “covid-19” should be written in capital letters (“COVID-19”) to homogenize the title to the text. If the editor agrees, the authors could consider the change of the title.

ABSTRACT

The abstract is well structured and properly reflects the main text highlighting the most important aspects of this paper. However, I recommend making a further revision of it to fix some small typing/language errors.

KEYWORDS

Authors did not correctly reported keywords from MeSH Browser. In particular, I checked for example “Convolutional Neural networks” and “Biomedical imaging” on MeSH Browser and this is not a KW. This is important, in my personal opinion, in order to increase the traceability of this paper (and consequently the possibility of the Journal to be cited by Readers and Stakeholders). I suggest the check of all KW. In addition, I believe there is a type and “and” before the last KW should be removed.

INTRODUCTION

Although the introduction fits the context of the study, it is concise. Sometime, many concepts clearly explicated in an exhaustive introduction could help readers to become passionate about reading the paper and using it as a reference.

In my opinion, it is very important to underline the role of CT in COVID-19 patients, especially those who were severely ill. In fact, it is true that chest CT is one of the main techniques to assess the severity of the pneumonic infection, allowing patients to be stratified into risk categories, estimate their prognosis, and help in medical decision making for hospitalization. However, CT also plays a pivotal role in the detection and monitoring of COVID-19-related complications, most of which are life-threatening, thus improving patient outcomes and overall survival. For example, it represents the best modality choice to indagate vascular alterations, both pulmonary and visceral, and can provide adequate information on the state of the gastrointestinal and renal systems, also allowing the evaluation of possible brain injuries. [Diagnostics (Basel). 2022 Mar 29;12(4):846. doi: 10.3390/diagnostics12040846  --  J Gastroenterol Hepatol. 2021;36(1):41. doi:10.1111/jgh.15094] Please, could the Authors cite this article and introduce these important aspects in this section?

Lines 43-44 “Since the outbreak of COVID-19, different methods to carry out the anal- 43 ysis of COVID-19 based on CT images have been increasing [18]-[52].” There are too many citations here for simple epidemiological facts, thus Authors should keep max 4-5 of them whereas the rest can be used in the remaining parts of the manuscript (since most of them are cited more than once afterwards).

Line 45 “Some works focus on the detection of infection regions [18]-[40].” Same consideration, please remove unnecessary citations.

Lines 45-108. This part is too long and becomes difficult to read. The authors must decide to reduce the text in order to increase reader’s engagement

MATERIALS & METHODS

Line 132. I believe there is a typo here, please check the sentence (remove “ Tello-Mijares and Woo”).

Line 189. Covid19 should be written in capital letters.

Lines 191-197. It is now well accepted in literature that there can be up to four categories for standardized reporting of chest CT findings related to COVID-19 were proposed by the RSNA Expert Consensus Statement [Radiol Cardiothorac Imaging. 2020 Jun 11;2(3):e200213. doi: 10.1148/ryct.2020200213]. Moreover, recent studies have demonstrated that this COVID-19 CT pattern prevalence did not statistically differ between the different waves of the pandemic and this new type of pneumonia tends to present itself with “typical” radiological features in the majority of cases [Emerg Radiol. 2021 Dec;28(6):1055-1061. doi: 10.1007/s10140-021-01937-y.] Could please the Authors discuss this topic here or in the Discussion section, citing the aforementioned papers? The CT images used for the present analysis were from COVID-19 patients with “typical” appearance?

In addition, could please the Authors state if all the included patients underwent RT-PCR confirmation for COVID-19? As the Authors correctly stated in the Introduction section, RT-PCR still is the principal technique for diagnosis of COVID-19 infection. [Radiology. 2020 Aug;296(2):E72-E78. doi: 10.1148/radiol.2020201160.]

RESULTS

Results remain clear and well-structured, and no further adjustments are needed.

DISCUSSION

This section is missing. As suggested in Tomography’s Instructions for Authors, Authors should discuss the results and how they can be interpreted from the perspective of previous studies and of the working hypotheses. The findings and their implications should be discussed in the broadest context possible. Future research directions may also be highlighted.

CONCLUSIONS

I think there is an error here because the Conclusion section because the Authors discuss about “thermographic breast images” with a “D CNN model” to “determine the difference between with cancer and without cases”. Since the previous sections are well-written and the topic is interesting, I want to believe there is a BIG TYPO. If the Authors could add an INHERENT conclusion to their manuscript, I think it would be considerable for publication.

TABLES & FIGURES

Tables & Figures are satisfactory, and they correctly match the quality standard of this Journal.

Author Response

Dear reviewer,  

Thank you very much for your contributions to improve our work. We improved all the paper, and we add the discussion and conclusions correctly. The paper was reviewed by someone who is an expert on English language and style.   

Point 1:

SPECIFIC COMMENTS

This manuscript is pleasurable and flowing; no major issues are appreciable.

Although language used is appropriate, I (I am not a native English speaker) recommend to Authors to obtain a certified native speaker with proficiencies in the scientific-medical field to complete properly this paper (if not jet done). Moreover, I recommend making a further revision of the manuscript to fix some small typing/language errors.

Response 1:

The paper was reviewed by someone who is an expert on English language and style. Thank you very much.    

Point 2:

TITLE

The title is clear and direct. Personally, I believe it could be improved and be more focused on results. “Saliency-preprocessing for covid-19 diagnosis on computed tomography by Deep Learning”. Moreover, I believe that “covid-19” should be written in capital letters (“COVID-19”) to homogenize the title to the text. If the editor agrees, the authors could consider the change of the title.

Response 2:

The title was changed and is now more focused on the result and "covid-19" was also changed to "COVID-19" to homogenize.

 Thank you very much for your recomendation.

Point 3:

ABSTRACT

The abstract is well structured and properly reflects the main text highlighting the most important aspects of this paper. However, I recommend making a further revision of it to fix some small typing/language errors.

Response 3:

Someone who is an expert on English language and style has changed the small typing and language errors. Thank you very much.   

Point 4:

KEYWORDS

Authors did not correctly reported keywords from MeSH Browser. In particular, I checked for example “Convolutional Neural networks” and “Biomedical imaging” on MeSH Browser and this is not a KW. This is important, in my personal opinion, in order to increase the traceability of this paper (and consequently the possibility of the Journal to be cited by Readers and Stakeholders). I suggest the check of all KW. In addition, I believe there is a type and “and” before the last KW should be removed.

Response 3:

The KWs have been changed according to your recommendation, now we use the ones from the Tomography magazine and MeSH, also the "and" has been removed. Thank you very much.  

Point 4:

INTRODUCTION

Although the introduction fits the context of the study, it is concise. Sometime, many concepts clearly explicated in an exhaustive introduction could help readers to become passionate about reading the paper and using it as a reference.

In my opinion, it is very important to underline the role of CT in COVID-19 patients, especially those who were severely ill. In fact, it is true that chest CT is one of the main techniques to assess the severity of the pneumonic infection, allowing patients to be stratified into risk categories, estimate their prognosis, and help in medical decision making for hospitalization. However, CT also plays a pivotal role in the detection and monitoring of COVID-19-related complications, most of which are life-threatening, thus improving patient outcomes and overall survival. For example, it represents the best modality choice to indagate vascular alterations, both pulmonary and visceral, and can provide adequate information on the state of the gastrointestinal and renal systems, also allowing the evaluation of possible brain injuries. [Diagnostics (Basel). 2022 Mar 29;12(4):846. doi: 10.3390/diagnostics12040846  --  J Gastroenterol Hepatol. 2021;36(1):41. doi:10.1111/jgh.15094] Please, could the Authors cite this article and introduce these important aspects in this section?

Lines 43-44 “Since the outbreak of COVID-19, different methods to carry out the anal- 43 ysis of COVID-19 based on CT images have been increasing [18]-[52].” There are too many citations here for simple epidemiological facts, thus Authors should keep max 4-5 of them whereas the rest can be used in the remaining parts of the manuscript (since most of them are cited more than once afterwards).

Line 45 “Some works focus on the detection of infection regions [18]-[40].” Same consideration, please remove unnecessary citations.

Lines 45-108. This part is too long and becomes difficult to read. The authors must decide to reduce the text in order to increase reader’s engagement

Response 4:

You are right that it is a comprehensive introduction, the introduction has already been improved. The importance of the role of CT in patients with COVID-19, especially those who were seriously ill, has been highlighted; works have now been cited [doi: 10.3390/diagnostics12040846; doi: 10.1111/jgh.15094] in the introduction to achieve this.

We now reduce the references of other works to 6 citations, now we only refer to those more related to our work. Thank you for helping us to increase reader’s engagement.

Thank you very much.

Point 5:

MATERIALS & METHODS

Line 132. I believe there is a typo here, please check the sentence (remove “ Tello-Mijares and Woo”).

Line 189. Covid19 should be written in capital letters.

Lines 191-197. It is now well accepted in literature that there can be up to four categories for standardized reporting of chest CT findings related to COVID-19 were proposed by the RSNA Expert Consensus Statement [Radiol Cardiothorac Imaging. 2020 Jun 11;2(3):e200213. doi: 10.1148/ryct.2020200213]. Moreover, recent studies have demonstrated that this COVID-19 CT pattern prevalence did not statistically differ between the different waves of the pandemic and this new type of pneumonia tends to present itself with “typical” radiological features in the majority of cases [Emerg Radiol. 2021 Dec;28(6):1055-1061. doi: 10.1007/s10140-021-01937-y.] Could please the Authors discuss this topic here or in the Discussion section, citing the aforementioned papers? The CT images used for the present analysis were from COVID-19 patients with “typical” appearance?

In addition, could please the Authors state if all the included patients underwent RT-PCR confirmation for COVID-19? As the Authors correctly stated in the Introduction section, RT-PCR still is the principal technique for diagnosis of COVID-19 infection. [Radiology. 2020 Aug;296(2):E72-E78. doi: 10.1148/radiol.2020201160.]

Response 5:

We remove “ Tello-Mijares and Woo”, and changed “Covid19” by “COVID-19”.

We improve the section as you suggest and add the references [doi: 10.1148/ryct.2020200213], [doi: 10.1007/s10140-021-01937-y.], [doi: 10.1148/radiol.2020201160.]

Now include and confirm that “all included patients underwent RT-PCR confirmation for COVID-19”.  

Thank you very much.

Point 6:

RESULTS

Results remain clear and well-structured, and no further adjustments are needed.

Response 6:

Thank you very much, we appreciate it too much.

 Point 7:

DISCUSSION

This section is missing. As suggested in Tomography’s Instructions for Authors, Authors should discuss the results and how they can be interpreted from the perspective of previous studies and of the working hypotheses. The findings and their implications should be discussed in the broadest context possible. Future research directions may also be highlighted.

Response 7:

Thank you, you are right, now we have added this section correctly according to your comments. We discuss the 6 previous studies mentioned in the Introduction.

Point 8:

CONCLUSIONS

I think there is an error here because the Conclusion section because the Authors discuss about “thermographic breast images” with a “D CNN model” to “determine the difference between with cancer and without cases”. Since the previous sections are well-written and the topic is interesting, I want to believe there is a BIG TYPO. If the Authors could add an INHERENT conclusion to their manuscript, I think it would be considerable for publication.

Response 8:

Now the big bug was fixed, thanks

Point 9:

TABLES & FIGURES

Tables & Figures are satisfactory, and they correctly match the quality standard of this Journal.

Response 9:

Again, thank you very much and we appreciate it too much.

Reviewer 2 Report

This article presented a work on classifying COVID-19 3-class disease-level from CT images. The topic is interesting but unfortunately, the presentation of the article is very concerning. Many details are missing in the methods, the results are not very well presented, the conclusion is completely wrong, not even related to this study, and the discussion section is missing. I suggest rejecting this article under the current presentation. If the authors consider resubmission in future, the following specific suggestions are provided for refinement.

  1. Introduction (lines 45–85 and lines 86-108): Within lines 45-85, the authors provided a throughout literature review of previous works on “detection of infection regions” from CT images via DL technique, which is appreciated. However, for a research paper, the introduction is supposed to support the motivation and the innovation of this present study. The authors evenly discussed each of the 23 references within this part, without providing an explicit interpretation to link them with the present study. It is not clear to me how these references are related to what the authors did in this study, how they were different from each other aside from they have different technical details, and what the unmet needs or unsolved problems in the field motivate the authors’ work. I also do not understand why this part is separated into five paragraphs, since they are listing even examples without a change of topic. Thus, I’d suggest the authors modify this part to provide more interpretations of the relationship between the present work and the previous literature, so emphasizing the significance and innovation. The same comment applies to lines 86-108.  
  2. Figure 1 and Figure 2 are overlapped in the layout.
  3. Materials and Methods (2.1): Even though the details of GGO-PI segmentation are in previous work, a brief description should be provided in the methods.
  4. Materials and Methods (2.2): Please provide the details of your CNN architecture (e.g., the number of layers and specific structure in each layer) and training-validation datasets.
  5. Experimental Results: The section 3.1 Dataset should be included in the “Materials and Methods”. Please provide brief information regarding the source of these data, for example, the institution(s) that these data were collected, what scanner(s) and acquisition protocol(s) were used to collect these data, and whether pre-processing and normalization were performed on the data for harmonization. Also, please indicate how the dataset is separated for training and validating CNN model.
  6. Experimental Results: Are the results presented in Figure 6 and Tables 1-2 from the training set or the validation/testing set? Also, I’d suggest the authors perform a ROC analysis to evaluate the performance.
  7. Experimental Results: Reporting three ways for calculating precision seems unnecessary. The authors commented the macro-average is preferable in this study, then why report the other two?
  8. Conclusion: The conclusion is not related to this presented study. The conclusion says the study is about “identifying breast cancer in thermographic breast images” but the work presented above is actually about lung CT and covid-19 diagnosis.

Author Response

Dear reviewer,  

Thank you very much for your contributions to improve our work. We improved all the paper, and we add the discussion and conclusions correctly. The paper was reviewed by someone who is an expert on English language and style.

Also, the title was changed and is now more focused on the result and "covid-19" was also changed to "COVID-19" to homogenize.    

Point 1:

This article presented a work on classifying COVID-19 3-class disease-level from CT images. The topic is interesting but unfortunately, the presentation of the article is very concerning. Many details are missing in the methods, the results are not very well presented, the conclusion is completely wrong, not even related to this study, and the discussion section is missing. I suggest rejecting this article under the current presentation. If the authors consider resubmission in future, the following specific suggestions are provided for refinement.

Response 1:

Thank you for your comments, based on them we have made the necessary corrections so that our work receives an opportunity

Point 2:

Introduction (lines 45–85 and lines 86-108): Within lines 45-85, the authors provided a throughout literature review of previous works on “detection of infection regions” from CT images via DL technique, which is appreciated. However, for a research paper, the introduction is supposed to support the motivation and the innovation of this present study. The authors evenly discussed each of the 23 references within this part, without providing an explicit interpretation to link them with the present study. It is not clear to me how these references are related to what the authors did in this study, how they were different from each other aside from they have different technical details, and what the unmet needs or unsolved problems in the field motivate the authors’ work. I also do not understand why this part is separated into five paragraphs, since they are listing even examples without a change of topic. Thus, I’d suggest the authors modify this part to provide more interpretations of the relationship between the present work and the previous literature, so emphasizing the significance and innovation. The same comment applies to lines 86-108. 

Response 2:

You are right that it is not a comprehensive introduction, the introduction has already been improved. The importance of the role of CT in patients with COVID-19, especially those who were seriously ill, has been highlighted; works have now been cited [doi: 10.3390/diagnostics12040846; doi: 10.1111/jgh.15094] in the introduction to achieve this.

We now reduce the references of other works to 6 citations, now we only refer to those more related to our work. Thank you for helping us to increase reader’s engagement.

Thank you very much for your recomendation.

Point 3:

Figure 1 and Figure 2 are overlapped in the layout.

Response 3:

The Figures have been arranged correctly. Thank you very much.   

Point 4:

Materials and Methods (2.1): Even though the details of GGO-PI segmentation are in previous work, a brief description should be provided in the methods.

Materials and Methods (2.2): Please provide the details of your CNN architecture (e.g., the number of layers and specific structure in each layer) and training-validation datasets.

Response 3:

You are right now we add the subsection "2.1 Pulmonary parenchymal identification by Poisson inverse gradient" and we also improve the explanation of obtaining the Ground-Glass Opacity and Pulmonary in subsection 2.2.

In section 2.3 we explain in more detail the CNN architecture and the training and validation data sets.

Thank you very much.  

Point 4:

Experimental Results: The section 3.1 Dataset should be included in the “Materials and Methods”. Please provide brief information regarding the source of these data, for example, the institution(s) that these data were collected, what scanner(s) and acquisition protocol(s) were used to collect these data, and whether pre-processing and normalization were performed on the data for harmonization. Also, please indicate how the dataset is separated for training and validating CNN model.

Response 4:

In materials and methods we now explain:

“The CT is taken with a Philips Brilliance 16-slice helical CT scanner (Koninklijke Philips, Eindhoven, NV, Nederland)”

“We used two-fold cross-validation, we randomly shuffle the dataset into two sets d0 and d1, so that both sets are the same size. We first train on d0 and validate on d1, followed by training on d1 and validation on d0.”

Thank you very much.

Point 5:

Experimental Results: Are the results presented in Figure 6 and Tables 1-2 from the training set or the validation/testing set? Also, I’d suggest the authors perform a ROC analysis to evaluate the performance.

Response 5:

Now we specify that we are using in Tables 1 and 2 and Figure 8 of results that we are using two-folds cross-validation. Thank you very much.

Point 6:

Experimental Results: Reporting three ways for calculating precision seems unnecessary. The authors commented the macro-average is preferable in this study, then why report the other two?

Response 6:

Thank you very much, we appreciate your observation. Now we only leave the macro-average analysis

Point 7:

Conclusion: The conclusion is not related to this presented study. The conclusion says the study is about “identifying breast cancer in thermographic breast images” but the work presented above is actually about lung CT and covid-19 diagnosis.

Response 7:

Apologies. Now the big bug was fixed, thanks a lot.

Reviewer 3 Report

The current article "Saliency-preprocessing for covid-19 diagnosis on computed tomography by Deep Learning" presents serious flaws. Therefore I would recommend rejection due to the following reasons:

a) The conclusion section relies on breast cancer findings, while the current article is developed based on COVID 19 data;

b) the article does not provide any discussion ;

c) The article requires complete native proofreading. Kindly correct the following lines: 88, 100, 109, 118; 

d) The figures 1 and 2 are overlapped, while figure 5 should highlight the procedure sequence; 

e) The studies presented in the introduction should be discussed and not merely highlighted to understand their novelty better. 

Author Response

Dear reviewer,  

Thank you very much for your contributions to improve our work. We improved all the paper, and we add the discussion and conclusions correctly. The paper was reviewed by someone who is an expert on English language and style.

Also, the title was changed and is now more focused on the result and "covid-19" was also changed to "COVID-19" to homogenize.    

Point 1:

The current article "Saliency-preprocessing for covid-19 diagnosis on computed tomography by Deep Learning" presents serious flaws. Therefore I would recommend rejection due to the following reasons:

Response 1:

Thank you for your comments, based on them we have made the necessary corrections so that our work receives an opportunity

Point 2:

a) The conclusion section relies on breast cancer findings, while the current article is developed based on COVID 19 data;

Response 2:

Apologies. Now the big bug was fixed, thanks a lot.

Point 3:

b) the article does not provide any discussion;

Response 3:

Thank you, you are right, now we have added this section correctly according to your comments. We discuss the 6 previous studies mentioned in the Introduction.     

Point 4:

c) The article requires complete native proofreading. Kindly correct the following lines: 88, 100, 109, 118;

Response 4:

We improved all the paper, and we add the discussion and conclusions correctly. The paper was reviewed by someone who is an expert on English language and style.

Thank you very much.  

Point 4:

d) The figures 1 and 2 are overlapped, while figure 5 should highlight the procedure sequence;

Response 4:

The Figures 1 nad 2 have been arranged correctly, and Figure 7 improve. Thank you very much.

Point 5:

e) The studies presented in the introduction should be discussed and not merely highlighted to understand their novelty better.

Response 5:

You are right that it is not a comprehensive introduction, the introduction has already been improved. The importance of the role of CT in patients with COVID-19, especially those who were seriously ill, has been highlighted; works have now been cited [doi: 10.3390/diagnostics12040846; doi: 10.1111/jgh.15094] in the introduction to achieve this.

We now reduce the references of other works to 6 citations, now we only refer to those more related to our work. Thank you for helping us to increase reader’s engagement.

Thank you very much.

Round 2

Reviewer 1 Report

thank you for your efforts.

Reviewer 2 Report

The authors have successfully addressed my previous comments and properly refined the manuscript. 

Reviewer 3 Report

Can be accepted now.